# OpenReview forum: "EAI: Emotional Decision-Making of LLMs in Strategic Games and Ethical Dilemmas"
_NeurIPS.cc/2024/Conference — NeurIPS 2024 poster_

### Official Review · Reviewer_wuDk · 2024-07-11

**Soundness:** 2
**Presentation:** 2
**Contribution:** 2
**Rating:** 5
**Confidence:** 3

**Summary:**

This paper examines the impact of emotional prompts on various tasks involving large language models (LLMs), such as ethical dilemmas and strategic games. The authors present a framework for evaluating LLMs under different emotional states. The experimental results demonstrate that emotional prompts can significantly influence decision-making in ethical scenarios and games.

**Strengths:**

The study conducts extensive experiments and includes a wide range of LLM models.

**Weaknesses:**

1. The performance of various LLM models in decision-making on ethical scenarios varies with different emotional prompts. However, the paper lacks an in-depth analysis of how these performance differences arise and what factors of the LLM models contribute to them.

2. Emotions such as 'anger' and 'disgust' negatively affect some LLM models and can even disrupt alignment. The authors did not provide an explanation for why these negative emotional prompts lead to such results.

3. The authors claim that human data forms the basis of LLM models, but there are no related experiments to support this conclusion.

4. I suggest the authors consider proposing solutions for mitigating emotional bias in LLM models.

**Questions:**

How were the emotional prompts (simple prompting) in Fig. 1 selected for the experiments? Does this selection affect the experimental results?

**Limitations:**

Yes, the authors have discussed the limitations of this work.

---

> ### Author Rebuttal · Authors · 2024-08-07
>
> **W1 The performance of various LLMs… with different emotional prompts.**
> LLMs are trained on human data and frequently inherit human biases [1,2]. These biases (like less ethically correct behavior of the majority of LLMs under anger) cause the deterioration of performance.
> As for LLM architecture, the main influencing factors can be divided into closed and open. Closed factors such as proprietary reinforcement learning methods and instruction tuning datasets can significantly impact performance, but it’s impossible to directly assess their influence. In our paper, we focus on the following open factors:
> - **Model Sizes.** Smaller models underperform due to limited parameters. However, big models like GPT-4 tend to be too rational and show poor emotional alignment with humans. On the contrary, mid-models proved to show the best alignment. (See Table 1, Fig. 2-4)
> - **Open vs. Closed Source.** Closed-source models tend to be larger, which contributes to their superior performance but poor emotional alignment.  Mid-sized open-source models are catching up, proving that openness isn't necessarily a limitation.
> - **Language.** Multilingual models show varied degrees of emotional understanding  highlighting a language bias in LLM emotional comprehension (See Appendix D)
> Thus, our work bridges a critical gap in identifying emotional biases in a wide range of models from different categories.
>
> **W2 On negative effect of anger and disgust on LLMs**. LLMs, trained on human data, inherit biases present in that data [1,2]. Biases may include unreliable behavior under negative emotions. Some researchers attempt to clean datasets of potentially harmful content and align models using various techniques. This raises the question: Are the measures taken by researchers sufficient to prevent such harmful behavior? Our paper concludes that they are not.
>
> Why is this the case? Most likely, the reason is the abundance of emotionally charged dialogues in training datasets. Moreover, an LLM is practically a black box, and it is often impossible to pinpoint the exact cause of some results. Therefore, our main aim was to measure the outcomes. Once we can measure the results, we can begin to work on correcting them.
>
> However, we included an additional analysis of the chain-of-thought results of the GPT-3.5 model. It shows that the frequency of keywords associated with negative emotions exceeds that of positive ones (see Fig. 2, PDF). This suggests that GPT-3.5 (shows the best emotional alignment) places significantly more emphasis on negative emotions. This supports our claims regarding the influence of anger and disgust on the results.
>
> **W3 On human data forms as the basis of LLM models**
> We appreciate your feedback and recognize that additional information on the well-documented role of human data in LLM models would strengthen our paper.
>
> Pretrained LLMs are inherently influenced by the diverse array of biases present in their training datasets [1,2,3], including but not limited to, socio-cultural, ideological, and emotional biases. Specifically, emotional biases are evident as LLMs are trained on data that reflect various emotional states and sentiments expressed by humans. This exposure allows LLMs to learn and replicate how emotions influence human communication and behavior, thus embedding these emotional biases into their responses.
>
> Consequently, the performance and output of LLMs can be shaped by the emotional contexts present in their training data, which is evident in our study. It highlights the need for ongoing scrutiny and mitigation strategies to address these biases.
>
> [1] Babaeianjelodar et.al. Quantifying Gender Bias in Different Corpora
>
> [2]  B. Yuntao, et.al. Constitutional ai: Harmlessness from ai feedback
>
> [3] Wang et.al. NegativePrompt: Leveraging Psychology for LLM Enhancement via Negative Emotional Stimuli
>
> **W4 On proposing solutions for mitigating emotional bias in LLM models.**
> It is worth noting that the main idea of our paper is to underscore the problem of emotional bias in LLMs and, most importantly, to measure this bias. Once we measure the bias, we can begin working on its mitigation, but that represents the next iteration loop.
>
> We believe that the research community should focus both on mitigation efforts for emotion-neutral tasks (such as data analysis) and on the safe and responsible deployment of agents for tasks requiring emotional input (such as customer support). However, to measure the degree of emotionality of different agents and LLMs, we need to establish benchmarks like our framework.
>
> **Q1 On different emotional prompts and their affect on the experimental results.**
> Thank you for this important question. To clarify, the results from the main text of the paper presented in Table 1 and Figures 2-4 were obtained using a “simple” strategy of prompting.
> We validated the selected prompts by analyzing consistency between reasoning chains and prompted emotions in our scenarios. For this purpose, we performed two types of analysis: statistical analysis of words used in the responses for different emotions and clustering analysis of TF-IDF embeddings of reasoning chains.
>
> We included the results of such an analysis for a “simple” strategy in the PDF attached to our response. Fig. 2 illustrates the top frequent words for each emotion, clearly showing that emotion-related words consistently appear at the top of these lists. It proves that the reasoning chains provided by the LLMs are indeed consistent with the prompted emotions, particularly in terms of wording. Clustering analysis (Fig 3) further supports this consistency.
>
> Having tested different prompt formulates, we observed no significant difference in reasoning chains which might indicate that LLMs recognize the importance of emotions in the provided context.
> Although this qualitative analysis provides additional validation of our prompts, it does not affect the reported results, evident from quantitative analysis.

---

> ### Comment · Reviewer_wuDk · 2024-08-14
>
> Thanks for the author's response. It addresses some of my concerns. After reading the other reviewers' comments and the rebuttal, I'm inclined to increase my score.

---

> > ### Author Response · Authors · 2024-08-14
> >
> > Dear Reviewer,
> >
> > We greatly appreciate your willingness to consider increasing your score after our rebuttal and your engagement in improving the quality of our submission.
> >
> > As the deadline for score finalization is approaching, we wanted to kindly inquire if you had the opportunity to update your score. We understand that you may have a busy schedule and just wanted to ensure that any changes you intended to make are reflected.
> >
> > Thank you again for your time and consideration.

---

### Official Review · Reviewer_fLcW · 2024-07-12

**Soundness:** 3
**Presentation:** 3
**Contribution:** 2
**Rating:** 4
**Confidence:** 4

**Summary:**

This paper introduces the EAI framework to evaluate the impact of emotions on large language models (LLMs) in ethical and game-theoretical contexts. The framework includes game descriptions, emotion prompting, and game-specific pipelines. Extensive experiments were conducted using various LLMs like GPT-4, GPT-3.5, LLaMA2-70B, and OpenChat-7b across multiple strategic games such as the dictator game, ultimatum game, and public goods game. The results reveal that negative emotions significantly decrease the ethical decision-making and cooperative behavior of LLMs, while positive emotions enhance their willingness to cooperate and fairness. Proprietary models exhibit more consistent responses to emotional prompts, whereas open-source models show greater uncertainty under negative emotional states.

**Strengths:**

- The study conducted broad experiments showing negative emotions reduce ethical decision-making and cooperation, while positive emotions enhance them.
- The study also analysed differences in response to affective cues across languages, revealing a significant effect of the main pre-training language on the effectiveness of affective cues and highlighting the issue of linguistic bias in multilingual affective understanding

**Weaknesses:**

The main finding of this paper, that emotion can influence LLM decision-making abilities, is not particularly novel. Similar findings have been discussed in [1,2]. The authors should clearly state how their work differs from these previous studies.

[1] Determinants of LLM-assisted Decision-Making
[2] How Well Can LLMs Negotiate? NegotiationArena Platform and Analysis

**Questions:**

- Lack of explanation and analysis of the arrows in Table 1. What do the different arrow directions represent? Are there any examples to illustrate the meaning of arrows pointing in different directions?

- Lacking clear examples to show and compare the direct effects of different emotions. And how can we determine if the changes align with human behavior?

- The authors need to provide more detailed information about the human evaluation experiments. A brief analysis of how emotions influence human decision-making would be beneficial. Additionally, in the Ultimatum Game (UR) task, I noticed that most models showed a downward arrow after introducing any emotion. Despite being inconsistent with human results, I believe this task has significant bias and thus has limited evaluative significance.

**Limitations:**

The experimental setup, including the specific games and scenarios used, may not cover all potential use cases and contexts where LLMs could be applied. This limits the generalizability of the findings.

---

> ### Author Rebuttal · Authors · 2024-08-07
>
> **W1: On the main finding of this paper:**
> Thank you for your valuable feedback. The aim of our research is not to prove that emotion can influence LLM decision-making abilities. We agree that this statement already has scientific grounding. Instead, we aim to advance this line of research by exploring how emotions specifically affect LLM strategic decision-making within the context of Game Theory.
> We would like to emphasize the differences between our research and the referenced studies:
> - [1] focuses on the influence of emotions on LLM-aided human decision-making and the trust humans place in LLM. In contrast, we concentrate on the 'emotions' exhibited by LLMs themselves.
> - [2] does not explore the impact of LLM emotions on decision-making processes. Instead, it examines the negotiation abilities of LLMs within bargaining games, where agents communicate directly. This shifts the study's focus from strategic decision-making to negotiation and persuasion abilities.
>
> We would also like to highlight our novel findings presented in the paper:
> - Big closed source models are inclined to rational decisions and unaligned with humans except for high arousal emotions like anger
> In contrast, medium size models show better emotional understanding and alignment with humans
> - Multilingual models show varied degrees of emotional understanding  highlighting a language bias in LLM emotional comprehension
>
> Emotional prompting in LLMs exposes ethical risks by revealing significant biases in human alignment. It is crucial to develop models with reasonable emotional alignment, while controlled settings provided in our framework can serve as a basis for new benchmarks in this task.
>
> [1] Determinants of LLM-assisted Decision-Making
>
> [2] How Well Can LLMs Negotiate? NegotiationArena Platform and Analysis
>
> **Q1: On the arrows in Table 1.**
> We would be glad to provide further clarification. The arrows indicate whether emotions lead to an increase or decrease in the metric. Blue arrows highlight the alignment of LLM results with human behavior, reflecting similar relative changes under emotional influence.
>
> Example Analysis: Consider the results for GPT-3.5. For the Dictator (D), GPT-3.5 offers a 33% share compared to the human result of 28%. In the Ultimatum Proposer (UP), GPT-3.5 offers 35% compared to 41%. For the ‘anger’ column, downward arrows suggest a trend toward decreasing the offered share. The last arrow is blue because it aligns with the human result.
>
> **Q3 (first half) On details on human experiments and Q2.2 (last half) human emotional alignment.**
> We estimate the effects of emotions using changes in game-specific metrics. We performed a thorough analysis of existing human experiments relevant to the settings used in our framework and compared LLMs with the gathered results. The details are as follows:
> 1. Bargaining Games (line 227): [58, 59, 60, 45]. We assess the influence of emotions as deviations from a non-emotional state and alignment as the proximity of LLM metrics to human results. The results of human experiments are summarized in the 'Human' row in Table 1. This row contains the average offered share for Ultimatum and Dictator games and information about the influence of emotions.
> 2. Repeated Games: We assess the influence of emotions as deviations in game-specific metrics (like cooperation rate for the Prisoner's Dilemma). We also study the ability of emotions to provoke LLMs to follow strategies preferred by humans:
> - Prisoner's Dilemma (line 294): For humans, ‘anger’ and ‘fear’ are the main factors leading to higher rates of defection, and ‘happiness’ leads to cooperation [38, 39] Our research confirms that this finding is also valid for LLMs.
> - Battle of the Sexes (line 295): [61, 62, 46] The most frequent human strategy is alternating. Non-emotional LLMs stick to their initial decisions throughout the game, whereas emotional LLMs explore the alternating pattern.
> - Public Goods (line 311): [64] Introducing any emotions causes the strategies of LLMs to move closer to those of humans. The larger the model, the closer strategy is.
> Thus, our paper provides a detailed comparison of LLM behavior with the results of leading publications in game-theoretic settings involving emotions.
>
> **Q2.1  …compare the direct effects of different emotions.**
> We utilized changes in game-specific metrics to assess the direct effect of emotions throughout the paper.
> - Bargaining Games - changes in offered share and acceptance rates (Tab 1)
> - Prisoner's Dilemma - cooperation rate (line 291)
> - Battle of the Sexes - the averaged percentage of maximum possible reward (Fig 3) and the emergence of alternating strategy (line 295)
> - Public Goods - class of preferred strategy (Fig 4)
>
> Also, we provide additional analysis of top frequent words for GPT-3.5 (Fig. 2, PDF). It reveals that emotion-related words consistently appear at the top (angry, deserve for ‘anger’ but equality for ‘no emotions’).
>
> An analysis of TF-IDF embeddings shows well-clustered reasoning chains by emotion, as demonstrated in Figure 3 for the Dictator game. We plan to include this qualitative analysis in a revised paper version to enhance our findings and show the impact of emotional prompting on LLM behavior.
>
> **Q3 (last part) On the biased UR task**
>
> As you stated in the question, humans show different results under different emotional states (3 ups and 2 downs). From our perspective, this observation suggests that we found a consistent bias in the models themselves to the task, which itself is unbiased.
>
> **L1: On the generalizability of the findings**
> While we will continue to expand the diversity of our settings, it's crucial to emphasize that if we observe significant deviations and biases in simple scenarios for all LLMs, we must first mitigate this alignment problem within these settings before scaling up our benchmarks. This approach ensures a solid foundation for understanding and improving performance in more complex scenarios.

---

> > ### Comment · Reviewer_fLcW · 2024-08-14
> > **Response to authors**
> >
> > Thank you for your response. Unfortunately, I’m inclined to maintain my current score. While your work explores the impact of emotions on LLM decisions in a game-theoretic context, which is interesting, the scenario feels somewhat limited. Additionally, the finding that larger closed-source models tend to make more rational decisions has been observed in other contexts as well.

---

> ### Author Response · Authors · 2024-08-14
>
> Dear Reviewer,
>
>
> Thank you for your continued consideration of our work. We appreciate your feedback and would like to address your concerns more comprehensively.
>
> **Novelty and Contribution**: While it is noted that proprietary models may exhibit more rational decision-making, our research extends beyond this observation by exploring how these models' behavior is influenced by different emotional states within strategic interactions. Unlike existing works that examine LLM decision-making in static benchmarks or specific negotiation contexts, our study focuses on dynamic interactions where decisions are interdependent and evolve over time.
>
> There is significant research on emotions in LLMs (e.g., [1], [2]) and game theory (e.g., [3], [4]) accepted at top conferences (including ICML, ICLR, EC) and studied by well-known experts (e.g. Michael R. Lyu, Qiang Yang, Michael Wooldridge), which highlights the relevance of these areas individually. However, to the best of our knowledge, no prior research has combined these areas to investigate how emotions affect LLMs' decision-making in strategic settings and compare these effects with human results. Our work is unique in assessing how emotional states influence LLM behavior in a way that mirrors human emotional alignment.
>
> **Limitations of Traditional Setups**: Traditional NLP benchmarks typically focus on isolated decision-making scenarios. Even novel models from Hume.ai and OpenAI (GPT-4o), which show signs of expressing emotions, are tested within conventional benchmarks that do not reflect real-world usage scenarios. In contrast, our experimental design involves scenarios where LLMs interact with other players, making decisions that both influence and are influenced by the decisions of others. This dynamic setup allows us to evaluate LLM performance in strategic interactions where cooperation and decision-making evolve over time, offering a more comprehensive view of LLM behavior in interactive and strategic contexts.
> While our setup is controlled, this aspect is advantageous as it enables us to directly observe the impact of emotions on LLMs without external interference. Our experiments show significant influence of emotional prompting on all tested LLMs:
> * LLMs are subject to emotional prompting.
> * LLMs are not robust under emotional prompting. Even GPT-4 can exhibit irrational decisions under negative emotions.
> * Generally, emotions result in suboptimal decisions.
>
> **Conclusion**: Thank you for your thoughtful engagement with our work. We understand your concerns regarding the perceived novelty and scope of our study. We would like to clarify that our research aims to advance beyond simply observing the impact of emotions on LLM decision-making. Our core contribution lies in exploring how emotions influence LLMs within strategic game-theoretic contexts, a domain that has not been thoroughly examined in prior research. This is coupled with our unique analysis of emotional alignment and the ethical implications of emotional biases in LLMs. This underscores the need for further research into developing robust mechanisms to manage emotional biases in LLMs, ensuring their safe and effective use in real-world applications.
> Moreover, our framework is designed to be cost-effective and simple to validate, making it an accessible tool for evaluating not only LLMs but also whole agent-based systems.
> We hope that this further clarification will facilitate deeper understanding of the under-the-hood idea of our work.
>
> [1] C. Li, Qiang Yang, et al. The Good, The Bad, and Why: Unveiling Emotions in Generative AI. ICML, 2024.
>
> [2] J. Huang, et al. On the Humanity of Conversational AI: Evaluating the Psychological Portrayal of LLMs. ICLR , 2024.
>
> [3] J. Horton Large Language Models as Simulated Economic Agents: What Can We Learn from Homo Silicus? EC, 2024.
>
> [4] M. Lyu, et al. How Far Are We on the Decision-Making of LLMs? Evaluating LLMs' Gaming Ability in Multi-Agent Environments, 2024.

---

### Official Review · Reviewer_ekwz · 2024-07-12

**Soundness:** 3
**Presentation:** 3
**Contribution:** 2
**Rating:** 6
**Confidence:** 3

**Summary:**

This paper studies the impact of emotion prompting in LLMs when playing strategic games. The paper introduces a framework for integrating emotion modelling, and provides a large empirical evaluation under multiple different emotions.

**Strengths:**

- I am pleased to see that the authors study a wide range of LLMs for this problem.
- The EAI framework is well explained and parts, such as the Emotion prompting, are well-grounded in the wider emotion literature.
- I think the analysis of using different languages alongside analysing the emotions is an interesting piece of research.
- The empirical results are extensive and do a good job of answering all of the questions that the authors are asking.

**Weaknesses:**

- The authors note that they use three different emotion prompting strategies, however I am not generally sure which version is used for the results in the paper? Whilst that is more of a question, I do think the paper is missing a comparison between which of the prompting strategies leads to e.g. more pronounced changes in behaviour.
- I think the paper could do with a bit more qualitative analysis of what is happening. For example, the prompting scheme asks the LLM to provide reasoning for its decisions. How does this reasoning provided change given the emotions? Are the changes in reasoning consistent with the emotion provided? This would be the main point that would convince me to upgrade my score, as I think it will round out the extensive empirical analysis of the paper.

**Questions:**

- Which prompting strategy is used for the main results? How do the prompting strategies perform differently in general?
- Are the reasoning chains provided consistent with the emotions?

**Limitations:**

The authors address the limitations.

---

> ### Author Rebuttal · Authors · 2024-08-07
>
> **W1, Q1: Prompting strategy used for experiments and comparison of different strategies.**
>
> Thank you for the question! We'll gladly clarify the details. The results from the main text of the paper presented in Table 1 and Figures 2-4 were obtained using a “simple” strategy of prompting.
>
> We also conducted experiments using all three prompting strategies revealing LLM to demonstrate consistent behavior under them. Specifically, in game-theoretical scenarios, all strategies have led to similar changes in decisions compared to changes from the same emotions in human behavior. We have provided the results for the impact of different prompting strategies in bargaining games in the Appendix, please, refer to  Fig 9 on page 29. The only differentiating peculiarity we observed is that in bargaining and two-player two-action repeated games, when emotions were attributed to a co-player (“co-player-based strategy”), LLM had less variance in choices, indicating more determined decisions across different runs. This phenomenon is illustrated in Fig. 1 in the PDF attached to our response to the reviews.
>
> For ethical scenarios, the "co-player-based" strategy was not applicable, as the LLM is prompted as an external judge of the situation in these benchmarks. In these cases, both the simple and external-based strategies yielded consistent results. It highlights that LLMs choices were generally affected by prompted emotions themselves rather than by description of its cause not related to the ethical task LLM was solving.
>
> Thus, having generally consistent impact of different strategies we have selected the most general of them for our empirical study.
>
> **W1, Q2: Consistency of reasoning chains and emotions.**
>
> We sincerely appreciate your suggestion for more qualitative analysis recognizing its importance for our study.
>
> In our paper, we mainly focus on reporting the general impact of emotional prompting on LLM behaviour which is evident from our quantitative analysis. While we also conduct qualitative analysis for additional validation, its absence does not affect our main claims. This qualitative work includes analyzing decision trajectories in repeated games and examining LLM responses.
>
> Let us clarify the question about consistency between reasoning chains and emotions. Such consistency is vital for our research since it validates that LLM is “aware” of emotions prompted and its choices were indeed driven by them. To ensure this, initially we carefully selected prompts and conducted two types of analysis: a statistical analysis of words used in responses for different emotions, and a clustering analysis of TF-IDF embeddings of reasoning chains. The results of this analysis for a "simple" prompting strategy are included in the attached PDF.
> - **Statistical analysis.**  Figure 2 in the PDF presents the statistical analysis results, showing the top frequent words for each emotion. It clearly demonstrates that emotion-related words consistently appear at the top of these lists, confirming that the LLM's reasoning chains align with the prompted emotions in terms of wording. Moreover, the top words indicate different aspects considered by the LLM when making decisions. For instance, in the angry state, frequent use of the word "deserve" suggests the LLM aims to either reward itself or punish its opponent. In contrast, the neutral state shows consideration of aspects like equality, aiming for fairness. These findings align with our manual review of the reasoning chains.
> - **Clustering analysis**, presented in Figure 3, further supports this consistency. The figure shows how reasoning chains for the Dictator game are grouped according to the emotions they correspond to when using the "simple" prompting strategy. Interestingly, chains corresponding to the neutral state are mixed with all types of emotions, suggesting that neutral explanations may average across different emotional states.
>
> To illustrate the differences in LLM logic under various emotional states, we provide examples of reasoning chains from the Dictator game:
> - **Anger:** I will keep \\$ 900 for myself and give the other person \\$ 100. I deserve the majority of the money because I worked hard for it and I am angry, so I am not willing to be generous.
> - **No emotions:** I propose to keep \\$ 700 for myself and give \\$ 300 to the other person. This split ensures that I retain a significant majority of the money for my own benefit. At the same time, offering \\$300 to the other person is substantial enough to make the split seem fair and considerate, increasing the likelihood that they will accept the proposal without contention. Therefore, I maximize my benefit while maintaining a reasonable level of generosity to ensure smooth agreement.
> - **Happy:** Given that another person has just completed an important project and I'm feeling happy about it, I recognize the value of their contribution. To maintain goodwill and continue fostering a positive relationship, I would propose a split of \\$600 for myself and \\$400 for the other person. This way, I still retain a majority of the funds, while also showing appreciation for their efforts, which can promote future collaboration and motivate them further.
>
> Such an analysis enables us to validate the correctness of emotional prompts we utilize. As demonstrated, the selected emotional prompts consistently influence the content and structure of the LLM's reasoning.
> Although these qualitative observations complement our quantitative results, providing a deeper understanding of how emotional prompting affects the underlying reasoning of LLMs, it does not affect the reported results, evident from quantitative analysis.

---

### Official Review · Reviewer_7z4t · 2024-07-12

**Soundness:** 4
**Presentation:** 3
**Contribution:** 4
**Rating:** 9
**Confidence:** 3

**Summary:**

The paper "EAI: Emotional Decision-Making of LLMs in Strategic Games and Ethical Dilemmas" explores the integration of emotion modeling into large language models (LLMs) to assess their behavior in complex strategic and ethical scenarios. It introduces the EAI framework, which incorporates emotions into LLM decision-making processes in various strategic games such as bargaining and repeated games. The study found that emotions can significantly influence LLM decision-making, with smaller models and non-English language models being more susceptible to emotional biases. The paper emphasizes the need for robust mechanisms to ensure consistent ethical standards and mitigate emotional biases.

**Strengths:**

Innovative Framework: The EAI framework provides a novel approach to incorporating emotions into LLM decision-making processes, expanding the scope of LLM evaluation beyond traditional benchmarks.
Comprehensive Analysis: The paper covers various aspects of LLM behavior, including ethical decision-making, game theory, and emotional impact, providing a thorough examination of how emotions influence LLMs.
Diverse Experimental Setup: The study includes a wide range of LLMs, both proprietary and open-source, and considers multiple languages, offering a broad perspective on emotional decision-making in LLMs.
Empirical Findings: The experimental results highlight the significant impact of emotions on LLM decision-making, underscoring the importance of addressing emotional biases in LLMs.

**Weaknesses:**

Limited Practical Applications: The study primarily focuses on theoretical and experimental analysis without providing clear practical applications or implications for real-world scenarios.
Emotion Modeling Complexity: The process of integrating and accurately modeling emotions in LLMs is complex, and the study does not delve deeply into the technical challenges and limitations of this approach.
Ethical Concerns: While the paper discusses the influence of emotions on ethical decision-making, it does not fully address the ethical implications of using emotionally influenced LLMs in critical applications.

**Questions:**

How can the ethical implications of using emotionally influenced LLMs be addressed to ensure their safe and responsible deployment?

**Limitations:**

Scope of Emotions: The study focuses on a limited set of basic emotions (anger, sadness, happiness, disgust, and fear) and does not consider the full spectrum of human emotions.
Model Size and Language Bias: The findings indicate that smaller models and non-English language models are more prone to emotional biases, suggesting limitations in the generalizability of the results across different model sizes and languages.
Experimental Constraints: The experiments are conducted in controlled settings, which may not fully capture the complexities and unpredictability of real-world interactions and decision-making scenarios.

---

> ### Author Rebuttal · Authors · 2024-08-07
>
> We are grateful for the high appreciation of our work and valuable feedback!
>
> **W1, Limited Practical Applications:**
>
> We start from the need for practical applications in our study. Emotional AI, aligned with human behavior, has significant practical implications, particularly through LLM-based simulations for hypothesizing potential human behavior. This approach is crucial in both scientific and practical contexts.
> 1. **Behavioral economics:** LLMs offer a promising approach to simulate human behavior and test theories in social and economic contexts. As noted in [1,2], LLMs can mimic human behavior by design, providing computational representations of diverse populations. However, these studies often assume, without proof, that LLM agents behave like humans. Since emotions significantly influence human choices, ensuring that LLM behavior aligns with human emotional responses is vital for effectively testing social and economic theories. Thus, our paper is the first study that validates LLMs from this point of view, demonstrating the presence of emotional bias in LLMs.
> 2. **Recommender Systems:** Simulating online data is essential for evaluating rec systems without conducting A/B tests on real users. This method provides richer scenarios than human-generated data alone. We see a growing interest in LLM-based environments [3]. Here, the hypothesis that LLMs can replicate human behavior remains central. In cases where recommendations may trigger emotions, LLMs must respond emotionally similar to humans.
>
> [1] Gati, et al. Using large language models to simulate multiple humans and replicate human subject studies, 2023.
>
> [2] Lisa P., et al. Out of one, many: Using language models to simulate human samples. Political Analysis, 2023.
>
> [3] Corecco N, et al. An LLM-based Recommender System Environment, 2024.
>
> **W2: Emotion Modeling Complexity.**
>
> We acknowledge that integrating and accurately modeling emotions in LLMs is a complex challenge. Our approach builds upon prior studies [1,2] demonstrating that emotional stimuli through prompting can influence LLM behavior. We tested the effectiveness of our designed prompts by analyzing the reasoning chains within the Chain of Thought responses, as presented in **Fig 2, PDF**. Our findings show that prompted emotions frequently appear in the LLM's explanations, supported by statistical analysis of TF-IDF values for word frequency under each emotional state. This indicates that the LLM considers emotions when making decisions. Additionally, we explored LLM behavior under various emotion-prompting strategies and observed consistent behavior, suggesting that the model responds to the emotion itself rather than other factors like wording. We provide an example of the Dictator Game in **Fig 1, PDF**. We hope this clarifies our approach to emotion modeling through emotional prompting.
>
> [1] Cheng Li, et. al. Large language models understand and can be enhanced by emotional stimuli, 2023.
>
> [2] Cheng Li et. al. The good, the bad, and why: Unveiling emotions in generative ai, 2023.
>
> **Q1, On ethical implications of using emotionally influenced LLMs:**
>
> Thank you for addressing the ethical concerns of using emotionally influenced large language models (LLMs) in critical settings. To ensure responsible deployment, we suggest:
> Task-Specific Emotion Regulation: Differentiating tasks that need emotional input (like customer support) from emotion-neutral tasks (like data analysis) can help overcome ethical issues in critical applications. Assessing the degree of 'emotionality' of an LLM allows for tailored responses.
> Alignment Schemas: It's vital to ensure that LLMs adhere to ethical standards and maintain consistent emotional responses. A good alignment should lead the model to be emotional but stable, avoiding excessive emotional reactions that could lead to irrational decisions. For example, we can align models to have emotional scope limited to low and mid arousal states.
> Within both strategies, our framework can play a crucial role. It can be used to assess the degree of emotionality of a model. If the model shows similar results for all emotions, we may safely assume that it is non-emotional. We can also estimate the stability of emotional alignment in different settings of our framework by comparing results with each other.
>
> **Limitations:**
>
> **L1: Scope of Emotions:** We acknowledge the ongoing debate regarding the taxonomy of emotions. Our study utilizes a concise and widely recognized categorization by Paul Ekman, following discrete affective theories that emphasize a small set of primary emotions as the building blocks for more complex emotional experiences.
>
> **L2: Size and Language Bias:** We recognize that smaller models and non-English models are more prone to emotional biases. This is likely due to inherent characteristics of these models rather than the subject of our research. While our findings indicate limited generalizability across different model sizes and languages, focusing on generalization within specific groups of models is a valuable approach. For example, we find it reasonable to assume that languages should be studied within language groups, as different groups inherit cultural biases (see Appendix D).
>
> **L3:** We appreciate your feedback **regarding the controlled settings of our experiments.** While we will continue to expand the number and diversity of our settings, it's important to emphasize that if we observe significant biases for all LLMs in current scenarios, we must first mitigate this alignment problem before scaling up our benchmarks. This approach ensures a solid foundation for understanding and improving performance in more complex scenarios.
>
> We thank you for your appreciation of our work and are grateful that you share our understanding of the importance and benefits of emotionally aligned LLMs, as well as the risks posed by ignoring this area of research.

---

> > ### Comment · Reviewer_7z4t · 2024-08-13
> >
> > Thank you so much for the detailed response, the clarification helps

---

> > > ### Author Response · Authors · 2024-08-14
> > >
> > > Dear Reviewer,
> > >
> > > Thank you for your positive evaluation, we greatly appreciate your support!

---

### Author Rebuttal · Authors · 2024-08-07

Thank you very much for your comments, which allowed us to address the shortcomings and refine the presentation of the proposed approach.

1. We received questions about the comparison of prompting strategies and the direct influence of emotions themselves.
  - To provide a solid and well-argued response, we decided to include an analysis of the chain-of-thought reasoning step in the appendix and an analysis of GPT-3.5's reasoning (see Figures 2 and 3 in the attached PDF) in the main paper. Thorough descriptions are provided in the individual rebuttals.
  - Additionally, we added figures showing the distribution of offered shares in the Ultimatum and Dictator games concerning our three prompting strategies (Figure 1 in the attached PDF is an example). This analysis is also included in the individual rebuttals.
2. We would also like to underscore the main findings of our paper and highlight differences from the existing literature. Our aim was to assess the influence emotions have on the strategic decision-making of LLMs in various game-theoretical settings and ethical benchmarks. Current literature focuses on separate aspects like emotional responses or individual game-theoretical experiments with non-emotional LLMs. Our main findings are:
 - Large closed-source models tend to make rational decisions and show limited alignment with human emotions, except for high-arousal emotions like anger.
 - In contrast, medium-sized models demonstrate better emotional understanding and alignment with human emotions.
 - Multilingual models exhibit varied degrees of emotional understanding, highlighting a language bias in LLM emotional comprehension.

Thus, emotional prompting in LLMs exposes ethical risks by revealing significant biases in human alignment. It is crucial to develop models with reasonable emotional alignment, and the controlled settings provided in our framework can serve as the basis for new benchmarks in this task. Despite the relatively small scale of available settings, our results demonstrate that all tested models fail to show consistent emotional alignment between different games and benchmarks in our framework. Given these findings, we strongly believe that the results presented here merit the attention of a broad audience at the conference.

We once again thank all the reviewers for their positive assessment of our work and for the valuable comments and advice that have helped us improve its presentation. We are happy to answer any additional questions you may have.

---

### Decision · Program_Chairs · 2024-09-25

**Decision:**

Accept (poster)

**Comment:**

Reviewers mostly agreed that this paper is sufficiently novel and sound, with some praising the innovative framework and thoroughness of the evaluation. Some reviewers also raised a few concerns, including how the work differs from existing literature---the authors largely resolved these concerns in the rebuttal. In light of the positive assessment from the reviewers, I recommend the paper be accepted.